# Neuropsychological Assessment of the Relationship of Working Memory with K-BIT Matrices and Vocabulary in Normal Development and ADHD Children and Adolescents

**DOI:** 10.3390/brainsci13111538

**Published:** 2023-10-31

**Authors:** Elena I. Rodriguez-Martínez, Raquel Muñoz-Pradas, Antonio Arjona, Brenda Y. Angulo-Ruiz, Vanesa Muñoz, Carlos M. Gómez

**Affiliations:** 1Developmental and Educational Psychology Department, University of Sevilla, 41018 Sevilla, Spain; elisroma@us.es; 2Human Psychobiology Laboratory, Experimental Psychology Department, University of Sevilla, 41018 Sevilla, Spain; raqmunpra@alum.us.es (R.M.-P.); aarjona@us.es (A.A.); bangulo@us.es (B.Y.A.-R.); lmunnoz@us.es (V.M.)

**Keywords:** ADHD, central executive, phonological loop, visuospatial sketchpad, K-BIT

## Abstract

Background: The present report tries to understand the possible relationship between working memory (WM) and intelligence measurements, using the direct scores of the Working Memory Test Battery for Children (WMTBC) and Kaufman’s Brief Intelligence Test (K-BIT), in normal development (ND) and diagnosed attention deficit hyperactivity disorder (ADHD) children and adolescents. Results: Partial correlations, discounting the effect of age, showed a significant correlation in ND subjects between the central executive (CE) component of WM and the WM visuospatial sketchpad (VSS) component and the WM phonological loop (PL); also, significant correlations were obtained for the WM VSS with the K-BIT Matrices scores, the WM PL with the K-BIT Vocabulary, and the K-BIT Matrices scores with the K-BIT Vocabulary. For ADHD subjects, there were significant correlations between WM VSS and WM CE, and WM VSS and K-BIT Matrices. We tested the robustness of these correlations by selecting a small number of subjects through permutations; a robust correlation between WM CE and WM PL in ND, and between WM VSS and WM CE and WM VSS and K-BIT Matrices scores was obtained. These results were also supported by mediation analysis. Conclusions: There is a relationship during development between WM as measured with WMTBC and general intelligence as measured with K-BIT in ND and ADHD subjects. The dysexecutive character of ADHD has been shown, given that by controlling for intelligence, the differences in WM performance between ND and ADHD disappear, except for WM CE. The results suggest that in ADHD subjects, the WM VSS component presents a more pivotal role during cognitive processing compared to ND subjects.

## 1. Background

Working memory (WM) is a psychological process that is responsible for the processes of retention, perception, and manipulation of information to make changes in behavior [1]. One of the most used models to understand the functioning of the WM is the model of Baddeley and Hitch [2]. This model explains that WM is made up of three components: phonological loop (PL), responsible for retaining verbal/auditory information; visuospatial sketchpad (VSS), responsible for the cognitive processing of visual and spatial contents; and central executive (CE), the main system to which the other two components are subordinated. The CE is responsible for attentional issues, problem-solving, planning complex strategies, and multitasking [3]. In addition, it covers other functions such as the coordination of verbal and visual information [4].

Concerning the structural development of the WM, the PL and VSS develop independently in the 5–8-year-old period [5]. The independence of the VSS and PL was confirmed in preadolescents (11–14 years old) [6]. A moderate association between the PL and CE was observed in children aged 6 and 7 [7]. Utilizing the Working Memory Test Battery for Children (WMTBC) [7], a working memory test derived from the Baddeley and Hitch [2] model, a confirmatory analysis demonstrated the independence of the three components of working memory in children and adolescents (aged 6 to 15 years) [4]. Despite the statistical independence between the three components, a closer association between CE and PL than between CE and VSS has been found. The WM reaches a plateau at 15 years, with CE attaining the highest performance later than the slave systems [8]. The increased efficiency of WM operations with age would permit the release of space and the increase in capacity which is observed in WM during development [9]. Although Baddeley and Hitch’s three-component model is highly prevalent, the update function is also an important WM operation that affects WM development [10,11].

The relationship between working memory and intelligence has been studied and found to be strongly related [12,13]. There is a correlation between the results of diverse WM tasks and intelligence tests [14]. Indeed, the memory index obtained from administering some subtests of the Wechsler Adult Intelligence Scale (WAIS) and the Wechsler Intelligence Scale for Children (WISC) makes it possible to assess the ability to attend to verbally presented information, manipulate it in immediate short-term memory, and then respond [15,16]. WM is therefore a component of these widely accepted intelligence tests.

The WM construct is statistically related to that of fluid intelligence [17,18] and is of particular importance in the development of curricular activities. According to Cowan [19], WM is a crucial part of cognitive processing, enabling the performance of more complex and necessary functions of daily life, such as language comprehension, reading, reasoning, or other executive functions. It also plays a key role in the interaction between received information and long-term memory in terms of retrieving stored information that may be relevant, reinforcing its role in the development of curricular activities [19]. Tourva et al. [14] developed a study to analyze the possible relationship between intelligence and three cognitive functions: attention, working memory, and processing speed. The results show that there is a statistically significant relationship between WM and general intelligence but not between the other two cognitive functions. More recently, however, the so-called watershed model based on structural equation modeling has shown that WM and processing speed play a mediating role between white matter development and fluid intelligence [20]. The relationship between working memory and fluid intelligence is important in educational settings, given the importance of WM in other school processes, such as mathematical performance [21,22,23] or reading comprehension [24,25].

The Kaufman’s Brief Intelligence Test (K-BIT) [26] is used in this report. This test measures general intelligence through two subtests. The first subtest measures verbal abilities through word recognition (vocabulary test) and non-verbal abilities through tests to complete analogies (Matrices test). Vocabulary is thought to be related to crystallized intelligence, while the Matrices test is more related to fluid intelligence [27]. Indeed, in a similar test (Raven’s Matrices) [28], it has been shown that fluid intelligence mediates through the dorsolateral prefrontal cortex and parietal areas in the control of interference in a WM task, suggesting a close relationship between these two cognitive functions [29]. The K-BIT Vocabulary subscale is related to vocabulary breadth and depth, as it is based on the recognition of simple images and finding the words associated with a definition. In this context, Gathercole et al. [30] argue that phonological memory plays a crucial role in early vocabulary development, a notion supported by research that extends its influence into middle childhood (6–11 years) in the context of developmental disorders [31,32,33,34]. In adolescence, the crucial importance of vocabulary breadth in understanding the relationship between working memory and language has been highlighted [35]. One of the advantages of using the K-BIT is that it takes much less time to administer than, for example, the more complete Wechsler Intelligence Scale for Children (WISC-V) [36], making it particularly useful for children and groups with attentional difficulties. Direct scores on intelligence tests increase with age into late adolescence and early adulthood, following a similar pattern to WM [37].

Attention deficit hyperactivity disorder (ADHD) is a pervasive developmental disorder that includes attention and/or impulsivity deficits. Poor success in curricular activities and performance in ADHD is associated with poorer adjustment in school, family, and various social contexts [38]. WM deficits have been associated with ADHD and are considered to be one of the main information processing deficits in ADHD, predicting impairments in more complex behavioral and cognitive processes [39,40,41,42]. Indeed, WM impairments in children with ADHD may be exclusively related to academic skills, unlike the other symptoms of the disorder [43]. Poorer performance has been observed in both phonological and visuospatial tasks [44,45], and it has even been possible to differentiate ADHD subtypes (combined and inattentive) according to CE impairment [46]. In terms of general intelligence, children with ADHD have been shown to have relatively lower IQs than normal development (ND) children [47,48]. This reduction in general intelligence is part of the normal phenotype of ADHD [49]. However, Cornoldi et al. [50] studied ADHD children with a high IQ and suggested that WM deficits in ADHD may be specifically mediated by deficits in attentional control, which would not be as evident in fluid intelligence tests that do not require a fixed time to be completed. Interestingly, the watershed model of fluid intelligence [20] showed that there is a differential relationship between certain white matter tracts and WM when tested in a community-ascertained sample, compared to an impaired developmental sample with problems in attention, learning, and/or memory. But also, the variability in the community-ascertained sample allowed WM and fluid intelligence to be separated, whereas in the impaired sample, both WM and fluid intelligence were not separated. Therefore, the relationship between WM and intelligence, although clearly related in ND, would present some specificities in developmentally impaired populations.

The aim of the present report is to show whether there is a relationship between WM, as measured with WMTBC [7], and fluid and crystallized intelligence, as measured by the K-BIT, in a developmental context. The hypotheses are that the executive components of WM will be related to the direct scores on the Matrices and Vocabulary scales of K-BIT and that VSS will be related to the Matrices scale, while PL will be related to the Vocabulary scale. A different pattern of relationships between WM and intelligence measures is expected in ND and ADHD children. The potential differential mediating role of WM in relation to the association between age and K-BIT measures in both ND and ADHD should reflect a comparable pattern to that of correlational analysis.

## 2. Methods

### 2.1. Participants

Two different samples participated in the present study: normal development (ND) and attention deficit hyperactivity disorder (ADHD) subjects. The ND sample is composed of 84 participants with corresponding ages between 6 and 18 years (M = 11.31, SD = 3.3, range: 6/18 years), including 45 males (53.5%) and 39 females (46.5%). The sample of ADHD subjects is composed of 36 subjects with corresponding ages between 6 and 17 years (M = 11.29, SD = 3.19, range:6/18), including 28 males (77.8%) and 8 females (22.2%). Age did not show significant differences between groups. In the ND group, controlled for age, no significant gender differences were obtained for any of the analyzed empirical variables after False Discovery Rate (FDR) correction [51]. In the ADHD group, controlled for age, only the vocabulary was higher in males than females (*p* = 0.045); however, the low number of females in this group raises doubts about the latter result. Given the low influence of gender and for the sake of simplicity, the gender factor will not be considered in the analysis.

Initially, the ADHD sample was composed of 41 children and adolescents. They were evaluated through two different methods: a structured interview and “Nesplora Aula” [52,53]. All participants had been prescribed different amounts of methylphenidate, but they did not take the medication 48 h before participating in the study. The parents filled out the DuPaul questionnaire [54]. After the evaluation, there were 5 cases with no diagnostic concordance between the questionnaire and the clinical interview, so these subjects were rejected for further assessment, and the ADHD sample was finally composed of 36 participants.

The participants had a medium socioeconomic status, and they were in the academic year that corresponds to their age. No psychological, psychiatric, or neurological disease was reported (except for the diagnosis of ADHD). When ADHD subjects were taking medication, it was suppressed 48 h before the experiment to carry out the assessment, and the parents or tutors of the participants signed a written consent (parents/tutors in the case of the children and adolescents) following the Helsinki protocol. The study was approved by the Bioethical Committees of Seville University and Jaen University.

### 2.2. Working Memory Test Battery for Children (WMTBC)

The participants were administered with the WMTBC [7]. This test is composed of 9 subtests, and by combining the results obtained in a specific order (phonological loop (PL) = test 1, 2, 3, 5; visuospatial sketchpad (VSS) = test 4, 8; central executive (CE) = test 6, 7, 9), the direct scores that correspond to the three major components of the WM (PL, VSS, and CE) were obtained. The order of administration was as follows:Digit Recall: The examiner speaks sequences of digits at a rate of one per second. The child recalls each sequence in the correct order.Word List Matching: The examiner speaks a sequence of words twice. The child decides whether the order of the words in the second sequence is the same as in the first. In some trials, the two lists are identical (the words are in the same order), in which case the child is asked to respond by saying “same”. In other trials, the word in the second list is in a different order to the first, more specifically, two adjacent words have been swapped in order. In these trials, the child should respond “differently”.Word List Recall: The examiner speaks sequences of short (two syllable) words for immediate recall. Each sequence must be recalled in exactly the same order as it was heard.Block Recall: The examiner taps sequences using the blocks on the block board. Initially, children are asked to recall the location of only one block. Then, sequences of two or more blocks are presented. It is important that each sequence is recalled in exactly the same order as was seen.Non-Word List Recall: The examiner speaks sequences of one-syllable non-sense words. The child recalls each sequence in the correct order.Listening Recall: The examiner reads a series of short sentences, some of which make sense and some of which do not. The child is asked to judge whether each sentence makes sense or not and to answer ‘true’ or ‘false’ accordingly. After the total number of sentences in a trial (ranging from 1 to 6) has been heard, the child is asked to recall the last word of each sentence in the order in which they were heard. It is important that each target word is recalled in exactly the same order as it was heard.Counting Recall: The examiner presents the child with arrays of dots to be counted on a series of cards. Once the dots have been counted, the child is asked to recall the total number of dots on each of the cards seen, in the order in which they were seen.Mazes Memory: The examiner traces a route through each maze by following a red line with his finger. Once the route has been demonstrated, the child is asked to recall it by drawing it in pencil in a Maze Memory answer booklet.Backward Digit Recall: The examiner speaks a sequence of digits for immediate recall. The child is asked to recall the list in reverse order, i.e., the recalled list should begin with the last digit heard and end with the first digit heard (e.g., “1, 2, 3” would become “3, 2, 1”).

The battery used was adapted from English to Spanish, following a series of criteria such as:-For the Word List Matching and Word List Recall subitems, disyllabic words were collected from a list of frequency of words used by children, with certain similar criteria such as the length between the original words and high frequency of use, to facilitate the completion of the test to younger age groups [55].-For the subtest of Non-Word List Recall, the same words were used as in the previous tests. To transform words into non-words, we changed the order of the vowels within the words and the order of the syllabi that made up each word.-For the subtest of Listening Recall, due to the characteristics of the subtest, the original sentences were translated from English to Spanish to keep the semantic meaning and the possible interference between words as in the original English sentences.

### 2.3. Kaufman Brief Intelligence Test (K-BIT)

K-BIT measures verbal and non-verbal intelligence in children, adolescents, and adults [26]. The test consists of two subtests, a Vocabulary test (verbal crystallized intelligence) and a Matrices test (non-verbal fluid intelligence). Finally, from the sum of both subtests, an overall IQ score is obtained. Here, only the subtests will be analyzed.

The order of testing K-BIT and WMTBC was counterbalanced. Due to difficulties in arranging the time schedules, the K-BIT was only administered to 21 ADHD subjects and the whole sample of ND subjects (84). The direct scores of the K-BIT for verbal (Vocabulary) and non-verbal (Matrices) in both groups were obtained.

### 2.4. Statistical Analysis

All the statistical analyses were computed with IBM SPSS Statistics 26.

#### 2.4.1. Mean Group Differences

Univariate ANOVAs of the groups as a between-subject factor were computed for the PL, VSS, and CE of the WMTBC, and Vocabulary and Matrices of the K-BIT, using age and gender as covariates. The obtained *p*-values were corrected for multiple comparisons using the False Discovery Rate (FDR) [51].

#### 2.4.2. Relationship between the Components of WM and K-BIT

To study a possible relationship between the components of the WM and the K-BIT components of intelligence, bivariate Spearman correlations were performed between the scores of each of the subtests. Additionally, the same correlations were computed, controlling for age. The obtained *p*-values were corrected for multiple comparisons using the FDR. The mediational analysis will be used to explore the possible role of WM in mediating between age and K-BIT scores. In this analysis, a series of coefficients quantify the relationship between the different variables (a: age and WM component; b: relates WM with K-BIT scores; a × b: quantifies the indirect effect of WM on the relationship between age and K-BIT; c’: indicates the effect of age on K-BIT independently of the WM mediation; c: is the direct relationship between age and K-BIT). This analysis was computed with JASP [56]. Only the a × b results, which correspond to the possible role of WM mediating in the relationship between age and K-BIT scores, will be reported, given the primary interest of the present report.

Given the different number of subjects in the ND (84 subjects for WMTBC and K-BIT) and ADHD (36 subjects for WMTBC and 21 subjects for K-BIT) groups, 10,000 correlations were computed with the selection of 17 subjects by permutation without replacement. This procedure will permit the elimination of the biases that would appear from the different numbers of subjects in ND and ADHD, while not discarding the information from any single subject. The values of the selected subjects during permutations were selected from the residuals of the regressions of the WM and K-BIT variables vs. age to minimize possible different ages of the ND and ADHD subjects. This method of selecting subjects permitted the validation of which correlations were not affected by the sample size and should be considered highly robust.

## 3. Results

The FDR-corrected ANOVA mean comparisons, with gender and age as covariates, showed significant differences between the groups’ PL (F(1,116) = 15.64, *p* < 0.001, η^2^ = 0.119), VSS (F(1,116) = 6.49, *p* = 0.012, η^2^ = 0.053), CE (F(1,116) = 28.08, *p* = 0.007, η^2^ = 0.195), Matrices (F(1,101) = 43.21, *p* < 0.001, η^2^ = 0.300), and Vocabulary (F(1,101) = 25.62, *p* < 0.001, η^2^ = 0.202). In these five direct scores of variables, ND presented significantly higher scores than ADHD. An additional univariate ANOVA to compare the WM direct scores of ND and ADHD, with the Matrices and Vocabulary covariates as additional covariates, showed that only the CE was statistically significant after FDR correction (F(1,99) = 7.55, *p* = 0.021, η^2^ = 0.071). K-BIT and WM direct scores were Spearman correlated with the impulsivity and inattention scores of the DuPaul questionnaire, but no significant correlations were obtained.

To study the age dependence of WM and intelligence in both samples, linear regressions of the direct scores of the three WM components and the two K-BIT components, with respect to age (measured in days), were computed. All the linear regressions were significant (Figure 1 and Figure 2) for ND, while for ADHD, all regressions were significant except for the PL. In all significant cases, there was an increase in direct scores with age. The obtained *p*-values were corrected for multiple comparisons using FDR.

To study a possible relationship between the components of WM and the K-BIT components, bivariate Spearman correlations were performed between the scores of each of the subtests in both samples (Table 1). For the ND sample, all tested FDR-corrected correlations were significant (see Table 1). The results obtained for the ADHD sample showed a statistically significant correlation between CE–PL; CE–VSS; CE–Matrix; VSS–Matrix; Matrix–Vocabulary.

To analyze the possible correlations between the components of WM and the components of intelligence, discounting age as a determining factor, partial correlations were computed between the direct scores of the subtests, controlling for the variable “age measured in days” (Table 2). For the ND subjects, significant correlations were obtained between CE–PL, CE–VSS, PL–Vocabulary, VSS–Matrix, and Vocabulary–Matrix. For ADHD subjects, significant correlations were obtained between CE–VSS and VSS–Matrix.

The mediation analysis showed that in ND, the a × b coefficients were significant for the following mediations: CE mediating between age and Vocabulary *p* = 0.012; CE for Matrices, *p* = 0.025; VSS for Matrices, *p* = 0.005; and PL for Vocabulary, *p* = 0.018. The mediation of VSS for Vocabulary and of PL for Matrices were not significant. For ADHD, the a × b significant effect was more limited, only the mediation of VSS between age and Matrices was significant, *p* = 0.044.

To test for the robustness of the obtained significant correlations and try to compute correlations with the same number of subjects in ND and ADHD, 17 subjects were selected from each sample without replacement and correlated 10,000 times. Figure 3 shows the *p*-values histogram for each correlation. For ND, only the correlation between CE–PL showed the most correlations obtained with the sampled subjects being significant, while for ADHD, CE–VSS and VSS–Matrices were highly significant.

## 4. Discussion

Although not specified as an aim, the results show statistically significant differences in direct scores between ND and ADHD children on the WM and K-BIT measures. This supports previous findings indicating poorer performance in ADHD children in WM [40] and general intelligence [49]. When controlling for K-BIT direct scores, only CE remains statistically significant, supporting that ADHD has features of a dysexecutive syndrome [40].

The first aim of this report was to show whether there is a relationship between WM as measured by the WMTBC, and general intelligence, as measured by the K-BIT, during development in two different samples: ND and ADHD subjects. Initially, it was hypothesized that the CE of WM would be related to the direct scores on the K-BIT Matrices and Vocabulary scales of K-BIT, while the VSS would be related to the Matrices scale and the PL would be related to the Vocabulary scale. These relationships would be more functionally associated if the age component was removed, as the common factor of maturation would be eliminated. The results obtained in the ND sample show that, as predicted, significant correlations were established between PL with Vocabulary and VSS with Matrices, and a trend for statistical significance was obtained for CE with Vocabulary and Matrices (FDR-corrected). These results suggest that WM is also related to other cognitive functions, such as intelligence. Previous findings seem to show that the short-term storage is mainly responsible for the relationship between intelligence and WM, and other cognitive processes, such as attention, are discarded as main protagonists [13]. These findings were supported after examining the relationship between three types of intelligence (general, fluid, and crystallized) and four cognitive functions (attention, processing speed, short-term storage, and WM). Tourva et al. [14] developed a study to analyze the possible relationship between intelligence and processing speed, attention, and WM. They showed that there was a statistically significant relationship between WM and general intelligence but not with attention and processing speed. The strong relationship between WM and intelligence has been widely replicated (see [57] for a review). The correlations obtained in the present report support such a claim and add a certain modular aspect, given the pattern of correlations obtained between WM and K-BIT (CE–Matrices; CE–Vocabulary, PL–Vocabulary, and VSS–Matrices). More recently, the role of WM as a mediator between white matter development and fluid intelligence has been established but also includes processing speed as an additional mediator [20]. The mediation analysis in ND subjects obtained in the present report supports the following idea: CE is a general mediator for the influence of age on intelligence as measured by K-BIT. This mediation becomes more modular when referring to verbal elements/items (PL for Vocabulary) and matrices (VSS for Matrices), suggesting the more general executive role of CE. The influence of WM on higher cognitive functions goes beyond its relationship with general intelligence, although it is mediated by intelligence. Research shows the importance of WM in other school processes, such as mathematical performance [21,22,23] and reading comprehension [24,25].

A positive correlation between WM components (CE–PL and CE–VSS) was also found in ND. Gathercole et al. [4] showed an independent development of the components, although a closer relationship was obtained with CE–PL than with CE–VSS. The present results also show that there is a statistically significant relationship between WM components, although CE and PL showed a closer relationship than CE and VSS, as previously demonstrated with a much larger sample [4,8].

With regard to ADHD, not only were lower WM and intelligence measures presented, as previously described [40,58], but in fact, a close relationship between intelligence and WM was shown; when controlling for intelligence, the differences in WM performance between ND and ADHD disappear, except for CE. Interestingly, the pattern of correlations obtained in the ADHD sample is very different from that of the ND sample, with the positive significant correlations being biased towards the visuospatial module both in the K-BIT and WM. Significant correlations, controlled for age and corrected for FDR, were obtained in the CE–VSS and VSS–Matrices. Interestingly, only the Matrices subtest was predicted by WM measures (VSS). This result would suggest that fluid intelligence (measured by the Matrices subtest) would be more connected to the WM components than crystallized intelligence (measured by the Vocabulary subtest) in the ADHD group [59]. These results suggest that the previously described pattern in ND of a high relationship between CE and PL breaks down in ADHD children, and the same happens with the Vocabulary measures. This could be due to a greater impairment of left hemisphere functional integration in ADHD compared to ND. This suggestion is also supported by the mediation analysis, which showed that only the VSS was a significant mediator in the relationship between age and K-BIT (VSS for Matrices). Indeed, the possibility that intelligence level and WM performance may not be directly related in ADHD has already been demonstrated [50]. However, some of the differences between ADHD and ND would be due to the different sample sizes of two groups, and the permutation analysis allowed the robustness of these correlations to be tested with a small sample size that was the same for all groups and tests.

Permutation analysis with a small resampled collection of subsamples showed a very robust pattern of correlations in WM CE–PL in ND, and CE–VSS and VSS–Matrices in ADHD. This finding supports a different pattern of interrelationships between cognitive processes in ND and ADHD, one that is more biased towards verbal control in ND and visually biased in ADHD. The small sample size in the ADHD group suggests that although permutation analysis indicates that the visually biased processing mode in ADHD appears to be statistically very robust, a large sample size may be needed to confirm the present results. It should be noted that the lack of robustness that appears in the ND correlations does not mean that these correlations are not present when a larger sample is tested, as described in Table 1 and Table 2 and discussed above.

In summary, the results in the ND correlations showed that the two intelligence subscales, Vocabulary and Matrices, were related to PL and VSS, respectively, suggesting a modular specific pattern in the relationship between WM and intelligence. In ADHD, however, the pattern of relationship between WM and intelligence (as measured by K-BIT) was limited to the relationship of CE with VSS, and VSS with Matrices, suggesting a more central role for visual information in cognitive processing. However, the results for ADHD should be treated with caution, given the small sample size of the ADHD group.

An important limitation of the present report is the wide range of ages of the subjects, which is due to recruitment limitations rather than to theoretical considerations, given that age differences have been found in the causal directionality of the WM PL vs. Vocabulary relationships [30] and in the strength of the white matter relationship with cognitive ability in the watershed model of fluid intelligence [20]. However, the linear relationship between most of the cognitive measures and age found in the present report suggests that although there may be transient changes in the level of the relationship between cognitive variables, they are unlikely to be sufficient to invalidate the conclusions of the present report in either group of subjects.

## 5. Conclusions

During the developmental process, an association is established between working memory (WM), as assessed by WMTBC, and general intelligence, as measured by K-BIT, in normally developing subjects and those with ADHD. The executive deficient nature of ADHD has been evidenced, as when adjusting for intelligence, disparities in working memory performance between typically developing individuals and those with ADHD disappear, except for the CE working memory component. The findings suggest that in individuals with ADHD, the VSS component of working memory plays a more crucial role during cognitive processing compared to typically developing individuals.

## Figures and Tables

**Figure 1 brainsci-13-01538-f001:**
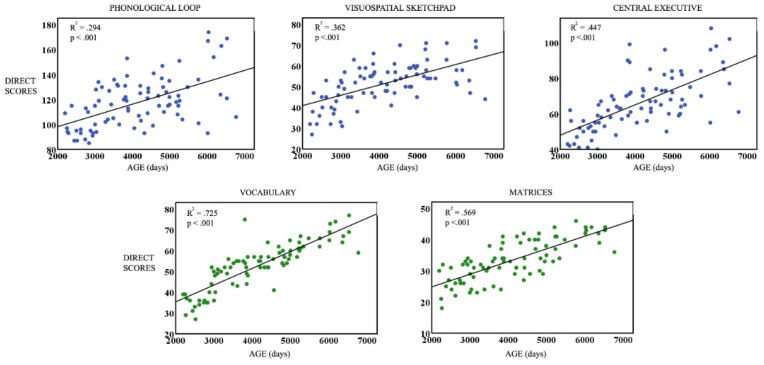
ND sample: Linear regressions of direct scores of the phonological loop, visuospatial sketchpad, central executive (WM components), Vocabulary, and Matrices (K-BIT components) vs. the age group measured in days. Blue dots represent scores for WM components, and green dots represents K-bit scores.

**Figure 2 brainsci-13-01538-f002:**
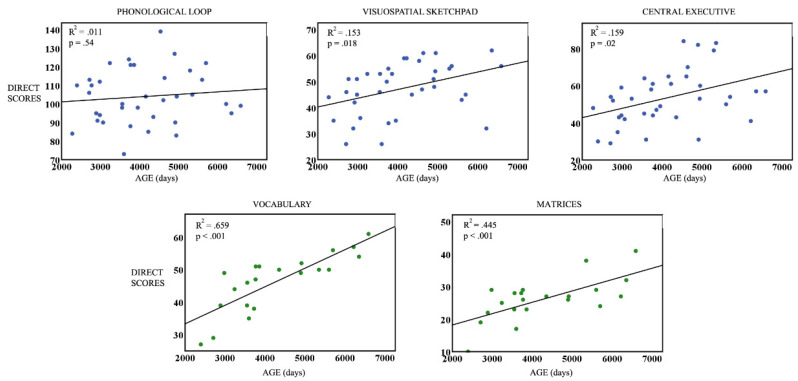
ADHD sample: Linear regressions of direct scores of the phonological loop, visuospatial sketchpad, central executive (WM components), Vocabulary, and Matrices (K-BIT components) vs. the age group measured in days. Blue dots represent scores for WM components, and green dots represents K-bit scores.

**Figure 3 brainsci-13-01538-f003:**
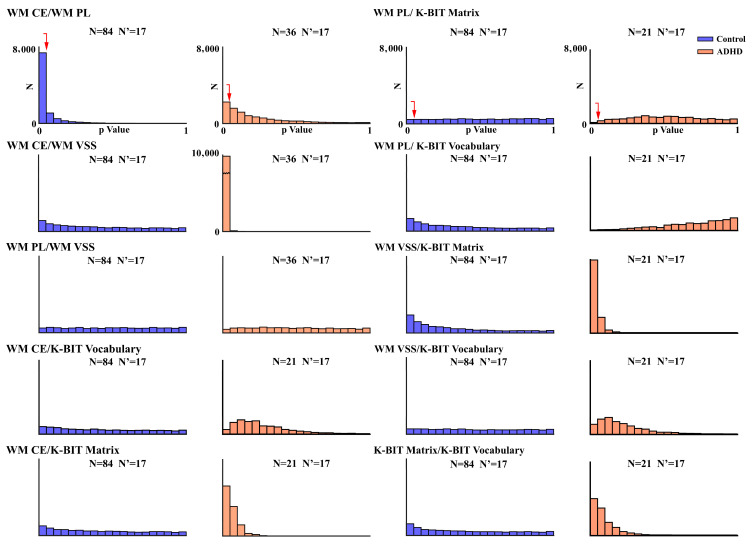
Probability values were obtained in the Spearman correlations for 17 subjects selected from 10,000 permutations without replacement. N: number of subjects of the original sample; N’: number of subjects selected for correlations from permutations. The red arrow indicates the *p* = 0.05 value. WM PL: phonological loop; WM VSS: visuospatial sketchpad; WM CE: central executive. Direct scores of the Vocabulary (K-BIT vocabulary) and Matrices subscales of the K-BIT (K-BIT matrix) are also indicated in the figure. Arrows indicate *p* = 0.05.

**Table 1 brainsci-13-01538-t001:** FDR-corrected Spearman correlations between the scores of each of the subtests of K-BIT and WMTBC in normal development (ND) and ADHD subjects. WM PL: phonological loop; WM VSS: visuospatial sketchpad; WM CE: central executive. Direct scores of the Vocabulary and Matrices subscales of the K-BIT are also indicated in the table. *p*-values were corrected using the (FDR). An asterisk indicates significant *p*-values.

Correlated Variables	ND			ADHD		
	R	*p*	N	R	*p*	N
WM CE/WM PL	0.753	<0.001 *	84	0.389	0.038 *	36
WM CE/WM VSS	0.581	<0.001 *	84	0.797	<0.001 *	36
WM CE/K-BIT Matrix	0.612	<0.001 *	84	0.644	0.005 *	21
WM CE/K-BIT Vocabulary	0.669	<0.001 *	84	0.261	0.361	21
WM PL/WM VSS	0.430	<0.001 *	84	0.162	0.432	36
WM PL/K-BIT Matrix	0.384	<0.001 *	84	−0.147	0.583	21
WM PL/K-BIT Vocabulary	0.503	<0.001 *	84	−0.001	0.997	21
WM VSS/K-BIT Matrix	0.587	<0.001 *	84	0.651	0.005 *	21
WM VSS/K-BIT Vocabulary	0.545	<0.001 *	84	0.349	0.201	21
K-BIT Matrix/K-BIT Vocabulary	0.706	<0.001 *	84	0.542	0.028 *	21

**Table 2 brainsci-13-01538-t002:** Partial Spearman correlations (controlling for age) between the scores of each of the subtests of K-BIT and WMTBC in normal development (ND) and ADHD subjects. WM PL: phonological loop; WM VSS: visuospatial sketchpad; WM CE: central executive. Direct scores of the Vocabulary and Matrices subscales of the K-BIT are also indicated in the table. *p*-values were corrected using the (FDR). An asterisk indicates significant *p*-values.

Correlated Variables	ND			ADHD		
	R	*p*	N	R	*p*	N
WM CE/WM PL	0.613	<0.001 *	84	0.368	0.066	36
WM CE/WM VSS	0.248	0.048 *	84	0.735	0.033 *	36
WM CE/K-BIT Vocabulary	0.220	0.063	84	0.317	0.231	21
WM CE/K-BIT Matrix	0.221	0.063	84	0.509	0.061	21
WM PL/WM VSS	0.101	0.403	84	0.177	0.376	36
WM PL/K-BIT Vocabulary	0.277	0.035 *	84	−0.044	0.849	21
WM PL/K-BIT Matrix	0.055	0.620	84	−0.178	0.489	21
WM VSS/K-BIT Vocabulary	0.175	0.138	84	0.353	0.194	21
WM VSS/K-BIT Matrix	0.335	0.009 *	84	0.551	0.048 *	21
K-BIT Vocabulary/K-BIT Matrix	0.245	0.048 *	84	0.466	0.066	21

## Data Availability

Data and code related to the present study are available under reasonable request from the corresponding author (cgomez@us.es).

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
