# Peer review of "Neuropsychological Assessment of the Relationship of Working Memory with K-BIT Matrices and Vocabulary in Normal Development and ADHD Children and Adolescents"

_brainsci, 2023, doi:10.3390/brainsci13111538_

Round 1

Reviewer 1 Report

Comments and Suggestions for Authors

The manuscript reports a relationship between WM and intelligence in typical and atypical development (ADHD). Overall, results show intelligence is dependent from WM development.
I found the data on the relationship intelligence-WM in typical development quite ascertained.
In my opinion, the most original achievement of the manuscript is the investigation of individuals with ADHD, where less research has been conducted.

I have a series of minor comments, listed below.

1. In the introduction, the state of the art about ADHD, working memory and intelligence is a little too narrow, I believe much more analysis of the literature is needed to warrant rationale and aims of the study.

2. The WM battery subtests should be briefly described (for instance, adding a few example items) as the reader can clearly understand which specific WM processes the labels refer to.

3. The age ranges considered are really wide. A theoretical assumption addressing this issue is missing, such as literature showing in both groups age-related differences are observed neither in intelligence nor in WM development. For instance, in typical development, it was found that phonological short-term memory exerts a direct causal influence on vocabulary acquisition (i.e., breadth), but only at early ages, not in later development. So, if the authors consider vocabulary as a measure of intelligence, they should discuss issues related to WM and vocabulary development, as these two processes clearly change during development. This issue is crucial to address as the rationale should include motivation to investigate such a wide age range.

Author Response

The manuscript reports a relationship between WM and intelligence in typical and atypical development (ADHD). Overall, results show intelligence is dependent from WM development.
I found the data on the relationship intelligence-WM in typical development quite ascertained.
In my opinion, the most original achievement of the manuscript is the investigation of individuals with ADHD, where less research has been conducted.

I have a series of minor comments, listed below.

1. In the introduction, the state of the art about ADHD, working memory and intelligence is a little too narrow, I believe much more analysis of the literature is needed to warrant rationale and aims of the study.

We have included in the manuscript new studies carried out by different authors that justify the study and the proposed objectives.

Attention Deficit Hyperactivity Disorder (ADHD) is a pervasive developmental disorder that includes attention, and/or impulsivity deficits. Poor success in curricular activities and performance in ADHD is associated with poorer adjustment in school, family, and various social contexts [38]. WM deficits have been associated with ADHD and are considered to be one of the main information processing deficits in ADHD, predicting impairments in more complex behavioral and cognitive processes [39-42]. Indeed, WM impairments in children with ADHD may be exclusively related to academic skills, unlike the other symptoms of the disorder [43]. Poorer performance has been observed in both phonological and visuospatial tasks [44; 45], and it has even been possible to differentiate ADHD subtypes (combined and inattentive) according to CE impairment [46]. In terms of general intelligence, children with ADHD have been shown to have a relatively lower IQs than normal development (ND) children [47, 48]. This reduction in general intelligence is part of the normal phenotype of ADHD [49]. However, [50] who studied ADHD children with high IQ, suggests that WM deficits in ADHD may be specifically mediated by deficits in attentional control, which would not be as evident in fluid intelligence tests that do not require a fixed time to be completed. Interestingly, the watershed model of fluid intelligence [20] showed that there is a differential relationship between certain white matter tracts and WM when tested in a community-ascertained sample compared to impaired developmental sample with problems in attention, learning and/or memory. But also, the variability in the community-ascertained sample also allowed WM and fluid intelligence to be separated, whereas in the impaired sample both WM and fluid intelligence were not separated. Therefore, the relationship between WM and intelligence, although clearly related in ND, would present some specificities in developmentally impaired populations.”

(This is in page 3)

2. The WM battery subtests should be briefly described (for instance, adding a few example items) as the reader can clearly understand which specific WM processes the labels refer to.

The following description of the subtests now appears in the text:

1. Digit Recall: The examiner speaks sequences of digits at a rate of one per second. The child recalls each sequence in the correct order.

2. Word List Matching: The examiner speaks a sequence of words twice. The child decides whether the order of the words in the second sequence is the same as in the first. In some trials the two lists are identical (the words are in the same order), in which case the child is asked to respond by saying "same". In other trials, the word in the second list is in a different order to the first, more specifically, two adjacent words have been swapped in order. In these trials, the child should respond "differently".

3. Word list recall: The examiner speaks sequences of short (two syllable) words for immediate recall. Each sequence must be recalled in exactly the same order as it was heard.

4. Block Recall: The examiner taps sequences using the blocks on the block board. Initially children are asked to recall the location of only one block. Then sequences of two or more blocks are presented. It is important that each sequence is recalled in exactly the same order as it was seen.

5. Non-word list recall: The examiner speaks sequences of one-syllable nonsense words. The child recalls each sequence in the correct order.

6. Listening Recall: The examiner reads a series of short sentences, some of which make sense and some of which do not. The child is asked to judge whether each sentence makes sense or not and to answer 'true' or 'false' accordingly. After the total number of sentences in a trial (ranging from 1 to 6) has been heard, the child is asked to recall the last word of each sentence in the order in which they were heard. It is important that each target word is recalled in exactly the same order as it was heard.

7. Counting Recall: The examiner presents the child with arrays of dots to be counted on a series of cards. Once the dots have been counted, the child is asked to recall the total number of dots on each of the cards seen, in the order in which they were seen.

8. Mazes Memory: The examiner traces a route through each maze by following a red line with his finger. Once the route has been demonstrated, the child is asked to recall it by drawing it in pencil in a Maze Memory answer booklet.

9. Backward digit recall: The examiner speaks a sequence of digits for immediate recall. The child is asked to recall the list in reverse order, i.e. the recalled list should begin with the last digit heard and end with the first digit heard (e.g. "1, 2, 3" would become "3, 2, 1")”.

(This is in pages 4 and 5.)

3. The age ranges considered are really wide. A theoretical assumption addressing this issue is missing, such as literature showing in both groups age-related differences are observed neither in intelligence nor in WM development. For instance, in typical development, it was found that phonological short-term memory exerts a direct causal influence on vocabulary acquisition (i.e., breadth), but only at early ages, not in later development. So, if the authors consider vocabulary as a measure of intelligence, they should discuss issues related to WM and vocabulary development, as these two processes clearly change during development. This issue is crucial to address as the rationale should include motivation to investigate such a wide age range.

We have included the justification in the text as follows:

In this context, Gathercole et al. [30] argue that phonological memory plays a crucial role in early vocabulary development, a notion supported by research that extends its influence into middle childhood (6-11 years) in the context of developmental disorders [31-34]. In adolescence, the crucial importance of vocabulary breadth in understanding the relationship between working memory and language has been highlighted [35].” (This is in pages 2 and 3.)

And also as you can see in the discussion, it is included as a limitation:

An important limitation of the present report is the wide range of ages of the subjects, which is due to recruitment limitations rather than to theoretical considerations, given that age differences have been found in the causal directionality of the WM PL vs Vocabulary relationships [30], and in the strength of the white matter relationship with cognitive ability in the watershed model of fluid intelligence [20]. However, the linear relationship between most of the cognitive measures and age found in the present report suggests that although there may be transient changes in the level of the relationship between cognitive variables, there are unlikely to be sufficient to invalidate the conclusions of the present report in either group of subjects.” (This is in page 11).

Reviewer 2 Report

Comments and Suggestions for Authors

This is a well-structured article. The main question addressed by this research is the neuropsychological assessment of the relationship of working memory with K-BIT matrices and vocabulary in children and adolescents with normal development and ADHD.

The introduction gives the background of this study as it briefly describes working memory, ADHD and the aims of the present study.

“Materials and Methods” section is descriptive enough. It refers to the participants, the neuropsychological batteries that were implemented and the statistical analysis carried out during this study.

The results are quite interesting and, to my opinion, well depicted in related figures and structured tables.

The discussion is well written, summarizing and discussing the main findings of the study and trying to associate it with recent relative literature data. The fact that the authors have also included a paragraph about the main limitations of the study, to my opinion, adds to the scientific value of this paper.

I believe that the authors should write a separate “conclusion” section summarizing their key findings and proposing some specific targets for future studies.

References, although relatively few, are relative to the subject.

Comments on the Quality of English Language

English language and style are generally fine. Only minor issues need to be addressed before publication. For example, some long sentences could be separated in shorter ones in order to make the text more comprehensible to the readers.

Author Response

This is a well-structured article. The main question addressed by this research is the neuropsychological assessment of the relationship of working memory with K-BIT matrices and vocabulary in children and adolescents with normal development and ADHD.

The introduction gives the background of this study as it briefly describes working memory, ADHD and the aims of the present study.

Materials and Methods” section is descriptive enough. It refers to the participants, the neuropsychological batteries that were implemented and the statistical analysis carried out during this study.

The results are quite interesting and, to my opinion, well depicted in related figures and structured tables.

The discussion is well written, summarizing and discussing the main findings of the study and trying to associate it with recent relative literature data. The fact that the authors have also included a paragraph about the main limitations of the study, to my opinion, adds to the scientific value of this paper.

I believe that the authors should write a separate “conclusion” section summarizing their key findings and proposing some specific targets for future studies.

References, although relatively few, are relative to the subject.

Thanks for your comments. We have added a specific section where the conclusions appear: 5. Conclusion

During the developmental process, an association is established between working memory (WM), as assessed by WMTBC, and general intelligence, as measured by K-BIT, in normally developing subjects and those with ADHD. The executive deficient nature of ADHD has been evidenced, as when adjusting for intelligence, disparities in working memory performance between typically developing individuals and those with ADHD disappear, except for EC working memory. The findings suggest that in individuals with ADHD, the VSS component of working memory plays a more crucial role during cognitive processing compared to typically developing individuals.” (This is in page 11)

Comments on the Quality of English Language

English language and style are generally fine. Only minor issues need to be addressed before publication. For example, some long sentences could be separated in shorter ones in order to make the text more comprehensible to the readers.

Thank you so much. We have revised the English throughout the manuscript and improved the text.

Reviewer 3 Report

Comments and Suggestions for Authors

This is an excellent study about ADHD and the importance of working memory. I would advice the authors to explain briefly the abbreviations, so that it can become clearer. The contents and the formal aspects are completely considered. The consequences of ADHD on specific forms of working memory have been pointed out through the statistical analysis and has been clarified very clearly. Therefor I think this is a very good study.

Comments on the Quality of English Language

This is an excellent study about ADHD and the importance of working memory. I would advice the authors to explain briefly the abbreviations, so that it can become clearer. The contents and the formal aspects are completely considered

Author Response

This is an excellent study about ADHD and the importance of working memory. I would advice the authors to explain briefly the abbreviations, so that it can become clearer. The contents and the formal aspects are completely considered. The consequences of ADHD on specific forms of working memory have been pointed out through the statistical analysis and has been clarified very clearly. Therefor I think this is a very good study.

Comments on the Quality of English Language

This is an excellent study about ADHD and the importance of working memory. I would advice the authors to explain briefly the abbreviations, so that it can become clearer. The contents and the formal aspects are completely considered

Thank you for your evaluation. We have reviewed and corrected all abbreviations in the document. We have excepted the description of the mean (M) and the standard deviation (SD) because they are well known.